# Rapamycin rejuvenates oral health in aging mice

Jonathan Y An[1,2], Kristopher A Kerns[1,3], Andrew Ouellette[4], Laura Robinson[4], H Douglas Morris[4], Catherine Kaczorowski[4], So-Il Park[2], Title Mekvanich[2], Alex Kang[2], Jeffrey S McLean[1,5], Timothy C Cox[6†], Matt Kaeberlein[1,2]*

[1]Department of Oral Health Sciences, University of Washington, Seattle, United States; [2]Department of Pathology, University of Washington, Seattle, United States; [3]Center of Excellence in Maternal and Child Health, University of Washington, Seattle, United States; [4]The Jackson Laboratory, Bar Harbor, United States; [5]Department of Periodontics, University of Washington, Seattle, United States; [6]Department of Pediatrics, University of Washington, Seattle Children's Research Institute, Seattle, United States

**Abstract** Periodontal disease is an age-associated disorder clinically defined by periodontal bone loss, inflammation of the specialized tissues that surround and support the tooth, and microbiome dysbiosis. Currently, there is no therapy for reversing periodontal disease, and treatment is generally restricted to preventive measures or tooth extraction. The FDA-approved drug rapamycin slows aging and extends lifespan in multiple organisms, including mice. Here, we demonstrate that short-term treatment with rapamycin rejuvenates the aged oral cavity of elderly mice, including regeneration of periodontal bone, attenuation of gingival and periodontal bone inflammation, and revertive shift of the oral microbiome toward a more youthful composition. This provides a geroscience strategy to potentially rejuvenate oral health and reverse periodontal disease in the elderly.

*For correspondence:
kaeber@uw.edu

Present address: †Department of Oral & Craniofacial Science, School of Dentistry, University of Missouri-Kansas City, Kansas City, United States

## Introduction

Old age is associated with failure to maintain homeostasis resulting in degradation of cellular maintenance and repair processes (*López-Otín et al., 2013*) and is the single greatest risk factor for many human diseases including cardiovascular disorders, dementias, diabetes, and most cancers (*Kennedy et al., 2014*). Interventions that target specific aging hallmarks have been shown to delay or prevent age-related disorders and extend lifespan in model organisms (*Kaeberlein et al., 2015*). Rapamycin, an FDA-approved drug, which directly inhibits the mechanistic target of rapamycin complex I (mTORC1), is one such intervention that extends lifespan and ameliorates a variety of age-related phenotypes (*Johnson et al., 2013*). In mice, rapamycin extends lifespan when administered beginning at 9 or 20 months of age (*Harrison et al., 2009*), and short-term treatments ranging from 6 to 12 weeks during adulthood have been shown to increase lifespan (*Bitto et al., 2016*), improve cardiac function (*Flynn et al., 2013*; *Dai et al., 2014*) and restore immune function as measured by vaccine response (*Chen et al., 2009*). Initial indications suggest that mTORC1 inhibition may also reverse declines in age-related heart function in companion dogs (*Urfer et al., 2017a*; *Urfer et al., 2017b*), and age-related immune function (*Mannick et al., 2014*; *Mannick et al., 2018*) and skin aging (*Chung et al., 2019*) in humans.

Periodontal disease is clinically defined by inflammation of the periodontium, the specialized tissue surrounding and supporting the tooth structure, resulting in clinical attachment loss, alveolar (periodontal) bone loss and periodontal pocketing, and pathogenic changes in the oral microbiome

**eLife digest** Age is the single greatest risk factor for many human diseases, including cancer, heart disease, and dementia. This is because, as the body ages, it becomes less able to repair itself. One way to prevent age-related disease and extend lifespan, at least in laboratory animals, is to use a drug called rapamycin. Mice treated with rapamycin live longer, have stronger hearts, and respond better to vaccination. But, despite these promising observations, the use of rapamycin as an anti-aging treatment is still under investigation. One open question is what age-related diseases rapamycin can help to prevent or treat.

In the United States, more than 60% of adults over the age of 65 have gum disease. These people are also more likely to have other age-related diseases, like heart disease or Alzheimer's. This association between gum problems and other age-related diseases prompted An et al. to ask whether it might be possible to treat gum disease by targeting aging.

To find out whether rapamycin could improve gum health, An et al. performed three-dimensional CT scans on mice as they aged to measure the bone around the teeth. Some of mice were treated with rapamycin, while the rest received a placebo. The mice that received the placebo started to show signs of gum disease as they aged, including inflammation and loss of bone around the teeth. The types of bacteria in their mouths also changed as they aged. Treating mice with rapamycin not only delayed the onset of these symptoms, but actually reversed them. After eight-weeks of the drug, the older mice had lost less bone and showed fewer signs of inflammation. There was also a shift in their mouth bacteria, restoring the balance of species back to those found in younger mice.

Rapamycin is already approved for use in people, so a clinical trial could reveal whether it has the same effects on gum health in humans as it does in mice. But there are still unanswered questions about how rapamycin affects the mouth as it ages. These include how the drug works at a molecular level, and how long the changes to gum health persist after treatment stops.

(*Könönen et al., 2019*; *Lang and Bartold, 2018*). Most recent epidemiologic data in the U.S. population suggests that more than 60% of adults aged 65 years and older have periodontitis (*Eke et al., 2012*; *Eke et al., 2015*), and diagnosis with periodontal disease is associated with increased risk for other age-related conditions including heart disease, diabetes, and Alzheimer's disease (*Gil-Montoya et al., 2015*; *Kim and Amar, 2006*; *Razak et al., 2014*).

Given that periodontal disease shows a similar age-related risk profile as other age-associated diseases (*An et al., 2018*), we predicted that interventions which target biological aging could be effective at treating periodontal disease. Consistent with that hypothesis, aged mice treated with rapamycin have greater levels of periodontal bone than control animals (*An et al., 2017*). In order to further test this idea and to understand potential mechanisms by which mTOR activity influences oral health during aging, we carried out a longitudinal study in which we asked whether transient rapamycin treatment during middle age can impact three clinically defining features of periodontal disease: loss of periodontal bone, inflammation of periodontal tissues, and pathogenic changes to the microbiome. Here we report that 8 weeks of treatment with rapamycin in aged mice is sufficient to regrow periodontal bone, reduce inflammation in both gingival tissue and periodontal bone, and revert the composition of the oral microbiome back toward a more youthful state.

## Results

In order to understand potential mechanisms by which aging and mTOR activity influence oral health, we carried out two parallel longitudinal studies at two sites in which aged mice were treated with either vehicle control or rapamycin for 8 weeks. NIA-UW mice were housed at the University of Washington and JAX mice were housed at The Jackson Laboratory (see Materials and methods). We used microCT (µCT) imaging to measure the amount of periodontal bone present in the maxilla and mandible of young (6 month), adult (13 month), and old (20 month) mice from the NIA-UW cohort (*Figure 1A*). The amount of periodontal bone for the maxilla and mandible of each animal was calculated as the distance from the cementoenamel junction (CEJ) to the alveolar bone crest (ABC) for 16 landmarked sites each on the buccal aspect of the maxillary and mandibular periodontium (*Figure 1B*). Thus, larger values represent greater bone loss. As expected, there was a significant

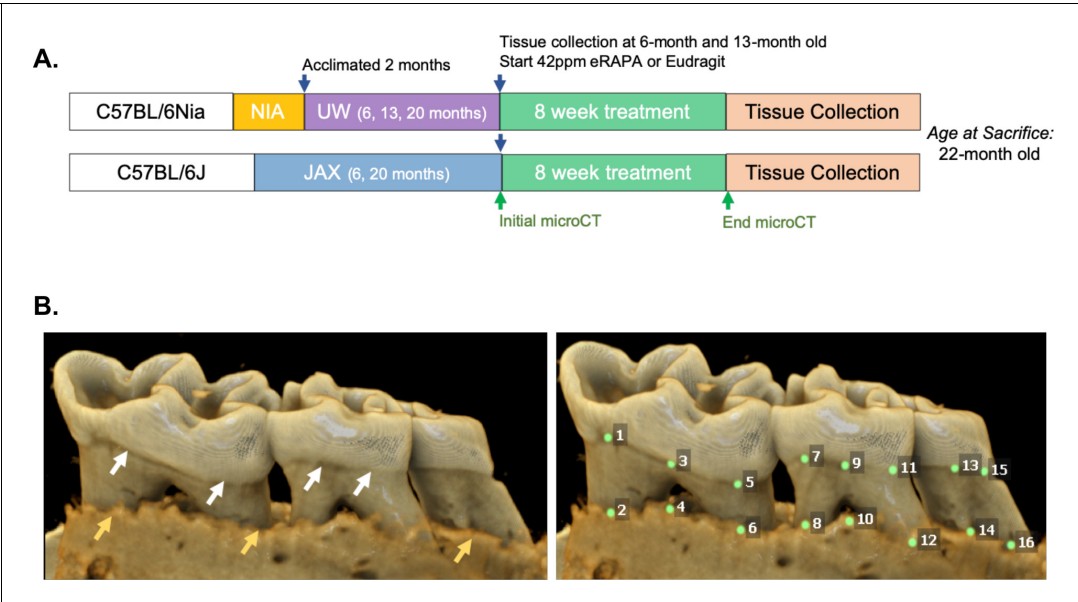

**Figure 1.** Cross-institution experimental design and assay for measuring periodontal bone loss. (**A**) The NIA-UW colonies were received directly from the NIA Aged Rodent Colony at 4, 11, and 18 months, then acclimated for two months within the UW facilities (ARCF) until they reached 6 (Young), 13 (Adult), and 20 months (Old). The Young and Adult cohorts were harvested for oral tissues and microbiome. The Old cohorts were randomized and either given Eudragit or 42ppm eRAPA within the food for 8 weeks. For the JAX colonies, an initial microCT image was taken prior to the 8 week treatment and then a final microCT before harvest. All animals were harvested at the end of 8 weeks, ~22 months old. (**B**) Representative image of a mandible is shown. Periodontal bone loss was measured as distance from the cementoenamel junction (CEJ, white arrows) to the alveolar bone crest (ABC, orange arrows) on 16 predetermined landmarks on the buccal aspect of maxillary and mandibular periodontium. The CEJ-ABC distances were totaled for each mouse.

loss of periodontal bone with age in the NIA-UW cohort (*Figure 2, A to C*). Mice treated with rapamycin for 8 weeks had significantly more bone at the end of the treatment period compared to mice that received the control diet (eudragit) (*Figure 2C*). To determine whether the increase in periodontal bone upon rapamycin treatment reflects attenuation of bone loss or growth of new bone, we performed µCT imaging on mice before and after treatment in the JAX cohort (*Figure 1A*). Old mice randomized into either the eudragit control or rapamycin treatment groups had significantly less periodontal bone than young mice prior to the treatment period (*Figure 2F*). After 8 weeks, the rapamycin treated mice had significantly more periodontal bone compared to eudragit controls and also compared to the pre-treatment levels for the same animals (*Figure 2, D to F*). The presence of new bone following rapamycin treatment can be observed by comparison of µCT images from the same animals before and after treatment (*Figure 2, D and E*).

Normal bone homeostasis results from a balance between new bone growth and bone resorption, which is reflected by the ratio of RANKL (receptor-activator of nuclear factor-κB ligand) to OPG (osteoprotegerin), and dysregulation of this balance contributes to bone loss in periodontitis (*Darveau, 2010*). Consistent with bone loss during aging, we detected significantly greater levels of RANKL in old animals of both cohorts compared to young animals (*Figure 3A and B*). OPG levels remained relatively stable, resulting in an increase in the RANKL:OPG ratio indicative of bone resorption exceeding bone formation (*Figure 3C*). These age-associated defects in bone homeostasis were suppressed by eight weeks of rapamycin treatment (*Figure 3*). In addition to increased RANKL:OPG ratio, a significant increase in TRAP[+] cells was also observed in periodontal bone with age (*Figure 3D and E*). TRAP (tartrate-resistant acid phosphatase) is a histochemical marker of bone resorbing osteoclasts (*Hayman, 2008*; *Ballanti et al., 1997*). Rapamycin treatment for eight weeks also decreased TRAP[+] cells. Together, our data indicate that rapamycin reverses periodontal bone loss in the aging murine oral cavity at least in part through inhibition of bone resorption.

Along with bone loss, gingival inflammation is a defining feature of periodontal disease. Aging is also associated with chronic accumulation of pro-inflammatory factors, a collective term referred to as inflammaging (*Chung et al., 2009*; *Franceschi and Ottaviani, 1997*; *Franceschi et al., 2000*;

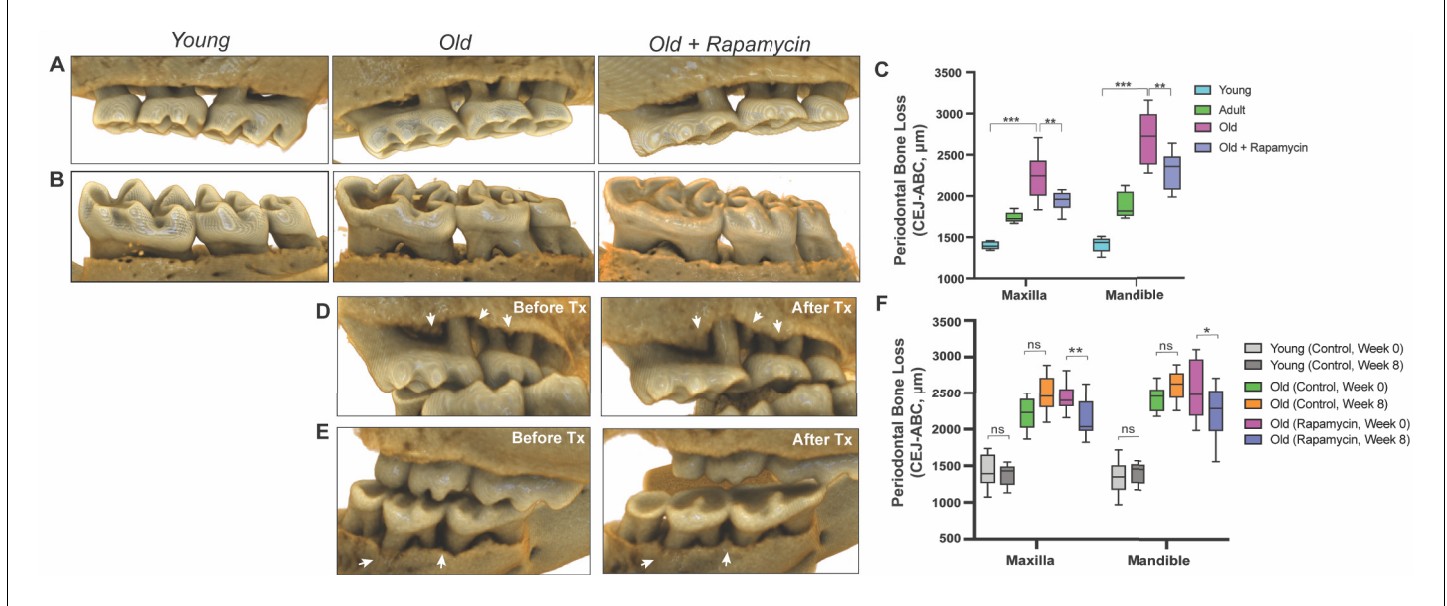

**Figure 2.** Rapamycin reverses age-associated periodontal bone loss (NIA-UW and JAX). (**A and B**) Representative images of NIA-UW (**A**) maxillary and (**B**) mandibular teeth of Young, Old, and Old treated with 42ppm eRAPA (rapamycin) revealing age-associated periodontal bone loss. 8 weeks of rapamycin attenuated periodontal bone loss. (**C**) Box-and-whiskers plots shows median, 25th and 75th percentile with whiskers at the 5th and 95th percentile. Statistical analysis was completed using unpaired t-test, with p-values <0.05 were considered statistically significant. *p<0.05, **p<0.01, ***p<0.005 (**D and E**) Representative images of the (**D**) maxillary and (**E**) mandibular teeth from the same animal in the JAX cohort before treatment (labeled Old) and after 8 weeks of 42ppm eRAPA (labeled Old+Rapamycin). On both the maxilla and mandible, there is periodontal bone loss around and in-between the molars, but after 8 weeks of 42ppm eRAPA the bone loss is reversed. White arrowheads indicate areas of bone loss and bone loss reversal (**F**) Box-and-whiskers plots shows median, 25th and 75th percentile with whiskers at the 5th and 95th percentile. Longitudinal comparison was completed with the same animal at baseline or after 8 weeks with either eudragit (control) or 42ppm eRAPA (rapamycin). Statistical analysis was completed using paired t-test, with p-values <0.05 were considered statistically significant. *p<0.05, **p<0.01, ***p<0.005.

*De Martinis et al., 2005*). The nuclear factor-κB (NF-κB) is a hub of immune and inflammatory response activated both during normal aging and as a consequence of periodontal disease (*Arabaci et al., 2010*; *Ambili and Janam, 2017*; *Abu-Amer, 2013*; *Liu et al., 2017*). We first evaluated the NF-κB hub through NF-κB p65 and IκBα expressions levels. The NF-κB heterodimer consists of RelA (or p65) and p50. IκBα functions as a negative regulator of NF-κB by sequestering it in the cytoplasm. Degradation of IκBα or phosphorylated-IκBα leads to nuclear localization of NF-κB subunits which induce expression of target inflammatory genes, such as TNF-α and IL-1β (*Liu et al., 2017*). In both the gingival tissue and periodontal bone, there was an increase in p65 expression with corresponding decrease of IκBα levels, indicating an age-associated increase in NF-κB inflammatory signaling in the periodontium (*Figure 4, A and B*). Eight weeks of rapamycin treatment was sufficient to reverse these changes. We also examined the levels of inflammatory cytokines in the oral cavity associated with normative aging and rapamycin treatment in mice. Consistent with the increase in NF-κB signaling, we found elevated expression of several cytokines in both the gingival tissue and the periodontal bone (*Figure 4, E and F*). Eight weeks of rapamycin treatment reversed most age-associated chemokine and cytokine changes in both the gingival tissue and periodontal bone. Thus, transient treatment with rapamycin during middle-age can largely restore a youthful inflammatory state in both the gingiva and periodontal bone of mice (*Wikham, 2016*).

Dysbiotic shifts in the oral microbiome are thought to play a significant role in the progression of periodontal disease in humans. We and others have previously shown that rapamycin treatment can remodel the gut microbiome in mice (*Bitto et al., 2016*; *Jung et al., 2016*; *Hurez et al., 2015*); however, the effect of rapamycin on the oral microbiome has not been explored. Therefore, we sought to evaluate effects of rapamycin on the aged oral microbiome using 16S rRNA gene sequencing and Amplicon Sequence Variant (ASV) analysis approach. Examination of the alpha diversity of the oral cavity illustrated a significant increase in species richness during aging that rapamycin

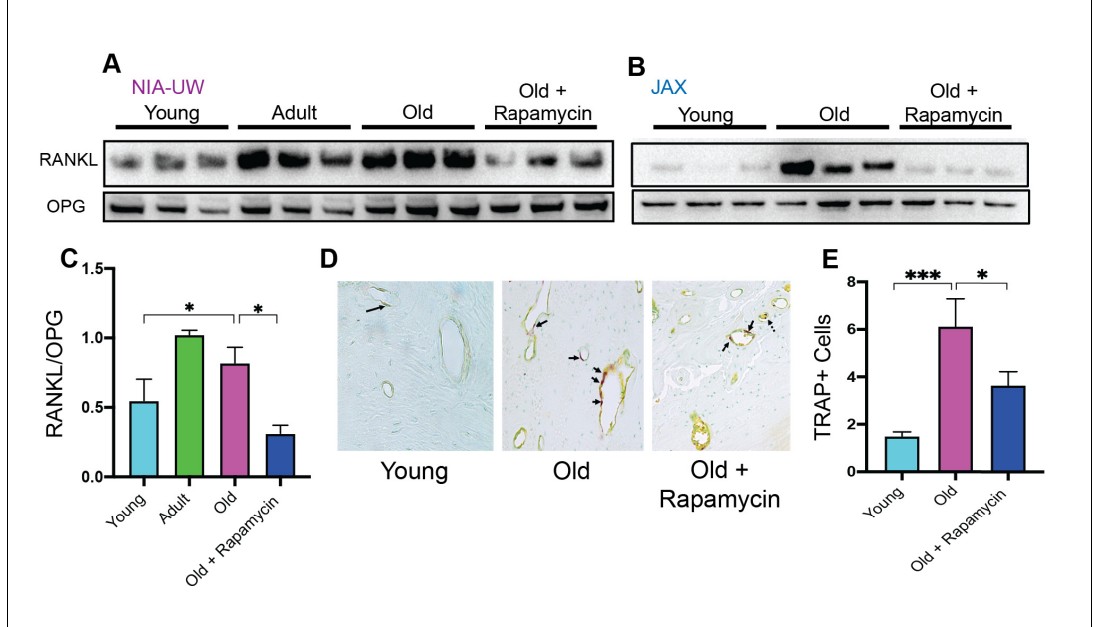

**Figure 3.** Rapamycin attenuates age-associated increase in RANKL expression and TRAP+ cells in periodontal bone. (**A and B**) RANKL and OPG expression was determined by western blot analysis of total lysates from the periodontal bone of aged animals (Young, Adult, and Old) and Old animals treated for 8 weeks with 42ppm rapamycin (eRAPA). The periodontal bone within both the NIA-UW and JAX Colonies showed an increased expression of RANKL while 8 weeks of rapamycin treatment ameliorated the increased RANKL expression. Each lane represents individual periodontal bone samples. (**C**) Quantification of RANKL/OPG of the NIA-UW western blot analysis. (**D**) Representative histological sections of the alveolar bone furcation that have undergone TRAP azo-dye staining with FastGreen counterstain. (**E**) Enumeration of TRAP+ cells within the periodontal bone from two-independent observers reveals an increase number of TRAP+ cells with age and diminishes with rapamycin treatment. Statistical analysis was completed with unpaired t-test, with significance set to p<0.05. *p<0.05, **p<0.01, ***p<0.005.

attenuated (*Figure 5A*, *Figure 5—figure supplement 1*). Among the most notable alterations in taxonomic abundance between groups was the reduction of *Bacteroidetes* phylum in the rapamycin-treated old animals (*Figure 5B*). When pooled across sites, no significant difference was observed between levels of *Bacteroidete*s in untreated young animals and old animals treated with rapamycin. Old animals treated with rapamycin in the JAX cohort had even lower levels of *Bacteroidetes* than young untreated mice (p<0.05), whereas in the UW cohort rapamycin treatment lowered the levels of *Bacteroidetes* to the level of the young mice (*Figure 5—figure supplement 2*). The *Bacteroidetes* phylum consists of over 7000 different species (*Thomas et al., 2011*) and includes bacteria associated with human periodontal disease such as *Porphyromonas gingivalis, Treponema denticola,* and *Bacteroides forsythus* (*van Winkelhoff et al., 2002*; *Torres et al., 2019*; *Socransky et al., 1998*). Further, both the *Firmicutes* and *Proteobacteria* phyla also showed a significant difference that was age dependent (*Figure 5B*) but was not significantly altered by rapamycin treatment. In order to assess whether rapamycin is shifting the composition back towards a youthful state, we evaluated the beta diversity using weighted UniFrac distances. We discovered a significant separation of the oral microbiome between old control and old rapamycin-treated animals, while no significant differences were observed between young mice and old rapamycin-treated mice (*Figure 5C*). Overall, we observed no significant differences in alpha diversity, beta diversity, nor relative taxonomic abundance between young untreated mice and old mice treated with rapamycin, suggesting an eight-week treatment with rapamycin reverted the old oral microbiome to a more youthful state. This observation is further supported when analysis of the samples is performed independently by facility (UW-NIA or JAX) (*Figure 5—figure supplement 3*). Despite differences in animal facility and diet composition, no batch effect was detected when comparing the JAX and NIA-UW cohorts (PERMANOVA, nperm = 999, p=0.34) (*Figure 5—figure supplement 4*).

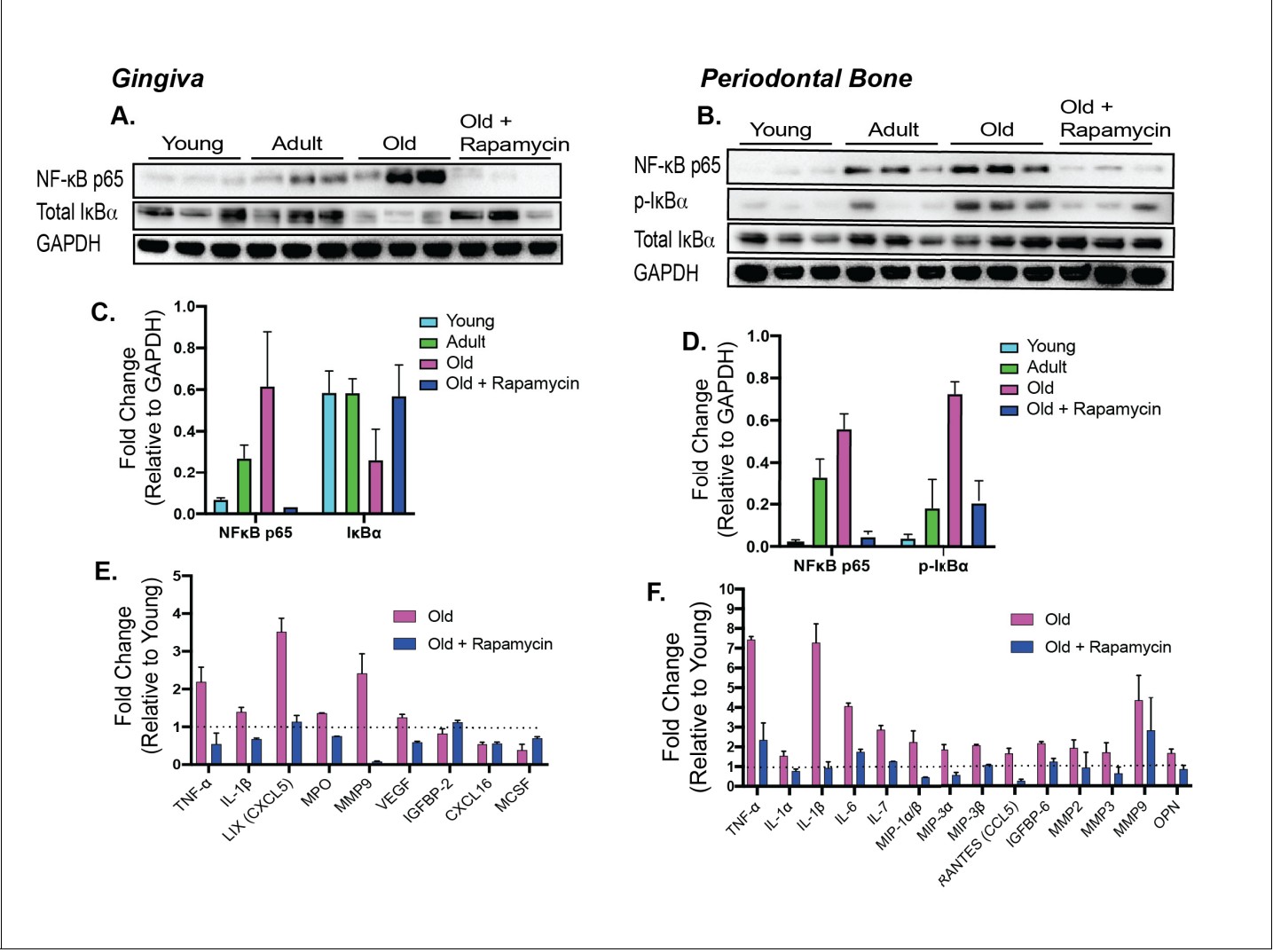

**Figure 4.** Rapamycin alters increased NF-κB expression and inflammatory cytokine profiles in periodontium. NF-κB p65 and IκBα expression was determined by western blot analysis of total lysates from the gingiva (**A,C**) and periodontal bone (**B,D**) of control animals (Young, Adult, and Old) and Old animals treated for 8 weeks with rapamycin (42ppm eRAPA). GAPDH was used a loading control. Both in the aging gingiva and periodontal bone, there is an overall increased expression of NF-κB p65 with corresponding alteration of IκBα or p-IκBα. 8 weeks of 42ppm eRAPA treatment attenuates the changes seen with age. For the gingiva, each lane represents gingiva from animals co-housed (n = 2), and each lane for the periodontal bone western blot represents individual animals. (**E and F**) Protein expression levels of mouse cytokines and chemokines was determined by a spotted nitrocellulose membrane assay (Proteome Profiler Mouse, R and D Systems) by loading pooled samples from (**E**) gingiva and (**F**) periodontal bone of Young and Old (Control, Eudragit), and Old animals treated for 8 weeks with rapamycin (42ppm eRAPA). Data are shown per manufacture's protocol, with fold-change relative to Young (Set to 1), expressed as mean ± SEM. All changes shown are statistically significant (p<0.05). CXCL16 and MCSF expression levels in (**E**) were not statistically significant.

## Discussion

Taken together, our data demonstrate that a short-term treatment with rapamycin in aged mice is sufficient to reverse three clinically defining features of periodontal disease: periodontal bone loss, periodontal inflammation, and pathogenic changes to the oral microbiome. This adds further support for the Geroscience Hypothesis, which posits that any intervention which targets the biological aging process will simultaneously delay multiple age-related diseases and functional declines (*Kaeberlein, 2017*; *Sierra and Kohanski, 2017*). To the best of our knowledge, this is the first report of rejuvenation in the aged oral cavity.

This work suggests several interesting questions that it will be important to evaluate in future studies. One such question is whether the effects of rapamycin on the aged periodontium will persist

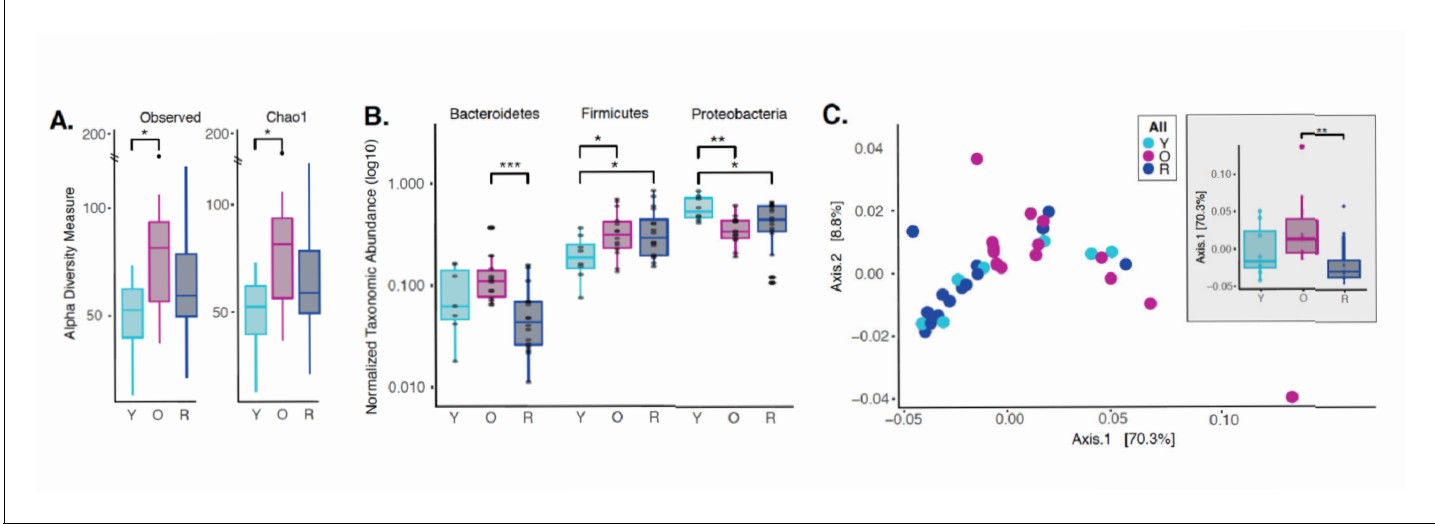

**Figure 5.** Rapamycin shifts aged oral microbiome towards young oral microbiome. (A) Alpha diversity for all samples reveal significant differences between young (Y) and old (O) mice without rapamycin treatment (p<0.05). (B) Phylum level abundance using normalized agglomerated data show significant difference for the *Bacteroidetes* (p<0.001) in old (O) mice and old mice with rapamycin treatment (R) for all samples. Also, significant changes are observed in the *Firmicutes* (p<0.05) and *Proteobacteria* phylum (p<0.05, p<0.01) that is age and treatment dependent. (C) Principal coordinate analysis using weighted Unifrac distances reveal beta diversity in the rapamycin-treated old (R) groups clustered with the young (Y). (C, inner panel) A significant separation between old (O) and rapamycin-treated old (R) groups (p<0.01; Axis 1, 70.3%) was observed, but no significant difference between young (Y) and rapamycin-treated old (R) groups was observed. *p<0.05, **p<0.01,***p<0.001.

The online version of this article includes the following figure supplement(s) for figure 5:

**Figure supplement 1.** Independent Alpha Diversity Analysis for JAX and UW-NIA animals.
**Figure supplement 2.** Independent phylum level abundance analysis for JAX and UW-NIA animals.
**Figure supplement 3.** Independent beta diversity for JAX and UW-NIA animals by principal coordinate analysis using weighted unifrac distances.
**Figure supplement 4.** Combined beta diversity for JAX and NIA-UW animals by principal coordinate analysis using weighted unifrac distances by site location and group designation.

after the treatment period or will rapidly revert back to the aged state. Improvements in age-related cardiac function associated with a similar rapamycin treatment regimen have been found to persist for at least eight weeks following cessation of treatment (*Quarles et al., 2020*), and it will be of interest to determine whether similar outcomes are observed for improvements in oral health. It will also be important in future studies to determine whether these effects are mediated through local inhibition of mTORC1 in the gingiva and periodontal bone or through systemic effects on immune function or other tissues. Likewise, it will be of interest to understand whether additional features of oral health that are known to decline with age, such as salivary function, are improved by rapamycin treatment. Finally, these results suggest the intriguing likelihood that additional geroscience interventions, such clearance of senescent cells, may phenocopy the effects of rapamycin in this context. Such interventions could pave the way for the first effective treatments to reverse periodontal disease and improve oral health in the elderly.

# Materials and methods

## Key resources table

| Reagent type (species) or resource | Designation | Source or reference | Identifiers | Additional information |
|---|---|---|---|---|
| Strain, strain background (*M. musculus*) | C57BL/6NIA | NIA Aged Rodent Colony | | PMID:27549339 |

*Continued on next page*

*Continued*

| Reagent type (species) or resource | Designation | Source or reference | Identifiers | Additional information |
|---|---|---|---|---|
| Strain, strain background (*M. musculus*) | C57BL/6J | Jackson Laboratory | RRID:IMSR_JAX:000664 | |
| Chemical compound, drug | Rapamycin | Rapamycin Holdings | | Amount based upon active rapamycin content to provide 42 parts per million concentration in chow. |
| Chemical compound, drug | Eudragit | Rapamycin Holdings | | |
| Chemical compound, drug | RIPA Lysis and Extraction Buffer | ThermoFisher Scientific | Cat#: 89901 | |
| Chemical compound, drug | HALT Protease Inhibitor Cocktail | ThermoFisher Scientific | Cat#: 78438 | |
| Chemical compound, drug | HALT Phosphatase Inhibitor Cocktail | ThermoFisher Scientific | Cat#: 78420 | |
| Chemical compound, drug | Restore Plus Stripping Buffer | Thermfisher Scientific | Cat#: 46430 | |
| Antibody | anti -NFkBp65 (Rabbit Monoclonal) | Cell Signaling | Cat#: 8242 | WB (1:1000) |
| Antibody | anti-phospho-IκBα (Mouse monoclonal) | Santa Cruz Biotechnology | Cat#: sc8404 | WB (1:1000) |
| Antibody | Anti- IκBα (Rabbit Monoclonal) | Abcam | Cat#: 32518 | WB (1:1000) |
| Antibody | anti-GAPDH (Rabbit monoclonal) | Cell Signaling | Cat#: 5174 | WB (1:1000) |
| Antibody | anti-RANKL (Mouse monoclonal) | Santa Cruz Biotechnology | Cat#: sc377079 | WB (1:1000) |
| Antibody | anti-OPG (Goat polyclonal) | R and D Systems | AF459 | WB (1:1000) |
| Antibody | anti-IgGκ (Mouse monoclonal) | Santa Cruz Biotechnology | Cat#: sc516102 | WB (1:10000) |
| Antibody | anti-rabbit IgG (Donkey polyclonal) | Thermo Fisher Scientific | Cat#: 31458 | WB (1:10000) |
| Antibody | anti-goat IgG (Donkey polyclonal) | Abcam | Cat#: ab97110 | WB (1:10000) |
| Commercial assay, kit | Proteome Profiler Mouse XL Cytokine Array | R and D Systems | Cat#: ARY028 | |
| Commercial assay, kit | Acid Phosphatase, Leukocyte (TRAP) Kit | Millipore Sigma | Cat#: 387A-1KT | |
| Commercial assay, kit | QIAamp DNA Microbiome Kit | Qiagen | Cat#: 51704 | |
| Commercial assay, kit | DNA Clean and Concentrator Kit | Zymo Research | Cat#: D4014 | |
| Commercial assay, kit | KAPA HiFi HotStart ReadyMix | KAPA Biosystems | Cat#: KK2601 | |
| Commercial assay, kit | Nextera XT Index Kit V2 | Illumina | Set A: FC-131–2001 Set D: FC-131–2004 | |
| Commercial assay, kit | AMPure XP Beads | Agencourt | A63881 | |
| Commercial assay, kit | SequalPrep Normalization Kit | Invitrogen | A1051001 | |

*Continued on next page*

*Continued*

| Reagent type (species) or resource | Designation | Source or reference | Identifiers | Additional information |
|---|---|---|---|---|
| Commercial assay, kit | TapeStation 4200 High Sensitivity D1000 assay | Agilent Technologies | G2991AA | |
| Commercial assay, kit | Tapestation Reagents | Agilent Technologies | 5067–5585 | |
| Commercial assay, kit | High Sensitivity D1000 ScreenTape | Agilent Technologies | 5067–5584 | |
| Commercial assay, kit | Qubit High Sensitivity dsDNA assay | ThermoFisher Scientific | Q32854 | |
| Commercial assay, kit | MiSeq Reagent Kit v3 (600 cycle) | Illumina | Cat#: MS-102–3003 | |
| Commercial assay, kit | PhiX Control Kit v3 | Illumina | Cat#: FC-110–3001 | |
| Software, Algorithm | Qiime2 | https://qiime2.org/ | V.2019.1 | |
| Software, Algorithm | DADA2 Package | PMID:27214047 | | |
| Software, Algorithm | Human Oral Microbiome Database (HOMD) | Homd.org PMID:30534599 | | v. 15.1 |
| Software, Algorithm | R-studio | https://rstudio.com/ | RRID:SCR_000432 | Version 3.5.3 |
| Software, Algorithm | Phyloseq | PMID:23630581 | RRID:SCR_013080 | |
| Software, Algorithm | Clustvis | PMID:25969447 | RRID:SCR_017133 | |
| Software, Algorithm | Ggplot2 | https://www.springer.com/gp/book/9780387981413 | RRID:SCR_014601 | https://www.springer.com/gp/book/9780387981413 |
| Software, Algorithm | Ampvis2 | http://dx.doi.org/10.1101/299537v1 | | |
| Software, Algorithm | vegan | https://cran.r-project.org, https://github.com/vegandevs/vegan | RRID:SCR_011950 | |
| Software, Algorithm | Ade4 | https://www.jstatsoft.org/article/view/v086i01 | | |
| Software, Algorithm | Bioinformatic scripts and microbiome data used in analysis | This paper | | https://github.com/kkerns85/Rapamycin_rejuvenates_oral_health_in_aging_mice.git. |
| Software, Algorithm | R Markdown | This paper | | https://rpubs.com/kkerns85/Rapamycin_Rmrkdown |
| Software, Algorithm | Graphpad Prism | Graphpad (graphpad.com) | RRID:SCR_002798 | Version 8.4 |

## Animal studies

To enhance rigor and reproducibility, experiments were performed on two different cohorts housed at two sites: the University of Washington in Seattle, WA and the Jackson Laboratory in Bar Harbor, ME. To examine the impact of rapamycin on the periodontium during normative aging, we designed a cross institutional study between the University of Washington (UW) and the Jackson Laboratory (JAX) (*Figure 1A*). The UW cohorts of C57BL/6Nia (hereafter termed NIA-UW Colony) were received directly from the National Institute on Aging (NIA) Aged Rodent Colony and acclimated within the UW facilities. The JAX cohorts of C57BL/6J (hereafter termed JAX Colony) were born and raised within the JAX facilities. We then treated mice at both sites with encapsulated rapamycin (eRAPA) in

the diet at 42ppm, which has been shown to significantly increase lifespan of UMHET3 and C57BL6/J mice (*Miller et al., 2014*; *Zhang et al., 2014*), or control food (eudragit). All data are from female mice, which have previously been found to have greater increases in lifespan and some health metrics compared to male mice at this dose of rapamycin (*Miller et al., 2014*; *Zhang et al., 2014*). For the NIA-UW colonies, five young, five adult, and 20 old (10 eudragit and 10 rapamycin) animals were utilized. While for the JAX colonies a total of 13 young and 26 old (13 eudragit and 13 rapamycin) animals were used. For this study, young, adult, and old mice were 6, 13, and 20 months of age, respectively.

## Seattle, WA

Twenty NIA-UW mice (10 on eudragit, 10 on rapamycin) received assigned diet treatments at 20 months of age, lasting for 8 weeks, along with five young and five adult mice as normative aging controls. Animals were housed individually in Allentown NexGen Caging (Allentown, Allentown, NJ) containing corncob bedding and nestlets. Mice were fed irradiated Picolab Rodent Diet 20 #5053 (Lab Diet, St. Louis, MO). Animals were maintained in a specific pathogen free facility within a *Helicobacter spp.*-free room. Mice were housed in groups and inspected daily. National Guidelines for the Care and Use of Animals and the IACUC guidelines were followed.

## Bar Harbor, ME

All methods are in accordance with The Jackson Laboratory Institutional Animal Care and Use Committee (IACUC)-approved protocols. Animals were fed standard Lab Diet 6% 5K52 with eRapa at 42 mg/kg/day or control. Animals had ad libitum access to food and water throughout the study. Animals were checked daily, and once per week the food was topped off. Animals were housed at 3–5 animals per cage.

A cohort of mice were transferred into the JAX Center for Biometric Analysis and brought into the imaging suite in groups of 10 mice per scan group. Prior to scanning, the weight of each mouse was recorded and anesthesia induced with 2–3% isoflurane. The mice were then placed in a prone position in the CT scanner and kept anesthetized for the duration of the scan with an isoflurane level of 1.2–1.5%. A whole head scan was performed with bone mineral density phantoms included on the specimen positioning bed. After the CT scan, the mouse was placed in a warmed isolation cage and allowed to fully recover from the anesthesia. At the end of the imaging session, the cohort was returned to animal housing facility.

Animal experimentation was performed in accordance with the recommendations in the Guide for the Care and Use of Laboratory Animals of the National Institutes of Health. All animals were handled according to approved institutional animal care and use committee (IACUC) protocols (#4359–01) of the University of Washington and (#06005-A24) of the Jackson Laboratory.

## Encapsulated rapamycin feeding

Encapsulated rapamycin (eRAPA) was obtained from Rapamycin Holdings, Inc. Food pellets were ground and mixed with encapsulated rapamycin at 42ppm. 300 ml of 1% agar melted in sterile water, and 200 ml of sterile distilled water were added per kilogram of powdered chow in order to make pellets. Pellets were stored at −20°C until use. Control food contained the same concentration of agar and encapsulated material (eudragit) without rapamycin at the concentration that matched the rapamycin chow. Eudragit is the encapsulation material used in eRAPA and is a copolymer derived from esters of acrylic and methacrylic acids. Eudragit without rapamycin was thus added to the regular chow at 42 ppm as a vehicle control.

## Micro-computed tomography (μCT) analysis

### Seattle Children's Research Institute and Friday Harbor Lab imaging parameter and processing

NIA-UW samples were scanned in a Skyscan 1076 and 1173 microCT system at the Small Animal Tomographic Analysis Facility (SANTA) at Seattle Children's Research Institute and Friday Harbor Laboratories at the University of Washington. Resolutions were 8–18 μm with following settings: 5 kV, 179μA, 360 ms exposure, 0.5 AI filter, 0.7° rotation step, and 3-frame averaging. Raw scan data were reconstructed with NRecon 1.6.9, and three-dimensional (3D) renderings were generated with

Drishti 2.7 (*Limaye, 2012*). For periodontal bone loss, 3D rendered images were randomized and landmarked by independent observers. Periodontal bone loss was measured as distance from the cementoenamel junction (CEJ) to the alveolar bone crest (ABC) on 16 predetermined landmarks on the buccal aspect of maxillary and mandibular periodontium. The CEJ-ABC distances were totaled for each mouse through the Drishti software, and means calculated. The analysis was completed by 3–4 independent observers.

### Jackson Laboratory (JAX) imaging parameter and processing

The mouse is scanned in a Perkin-Elmer Quantum GX in vivo Micro-CT tomograph. Resolutions were 17–50 microns with the following settings: 55 kV, 145 μA, 4 min exposure over 360 degrees rotation. The native Perkin-Elmer Viewer VOX image files are converted to Drishti Volume Exploration and Presentation Tool NetCDF format volumes using custom code specific for this study (*Limaye, 2012*).

## Western blot and proteome profile analysis

For protein analysis by western blot, gingival tissue and alveolar bone was dissected. Total cellular proteins were extracted in RIPA Lysis and Extraction Buffer (Thermo Scientific, MA, USA) and EDTA-free Halt protease and phosphatase inhibitor cocktail included to prevent protein degradation during extraction process. Gingival tissue was pooled from co-housed animals and bone samples were from single specimens. Protein concentration was determined by Pierce BCA Protein Assay Kit (Thermo Scientific). 10–20 μg of total protein was separated by SDS-PAGE on 10% or 12% (w/v) polyacrylamide gel, then transferred to PVDF membrane using Trans-Blot Turbo Transfer System (Bio-Rad, CA, USA). Antibodies to NF-κB p65 (D14E12) XP (8242, Cell Signaling Technology), phospho-IκBα (B-9, Santa Cruz), IκBα (32518, Abcam), GAPDH (D16H11) XP (5174, Cell Signaling Technology), RANKL (G-1, sc377079, Santa Cruz), and Mouse OPG (R and D Systems, AF459) were used to probe the membrane. Dependent upon the strength of the antibody-dependent signal, either the membranes were stripped with Restore Plus Western Blot Stripping Buffer and reprobed for total antibody, or duplicate gels were run and separate blots probed.

Analysis of the cytokine proteome was completed using a Mouse XL Cytokine Array Kit (R and D Systems, Bio-Techne Corporation, MN, USA). Gingiva and alveolar bone samples were individually pooled, protein concentration determined by Pierce BCA Assay Kit and 200 μg of protein lysate loaded. Detection and imaging were performed using ChemiDoc XRS+ (Biorad, USA) and Image Lab Software (Biorad, USA). Data analysis was completed per the manufacture's protocol.

## Histology

Tissues were fixed in Bouin's solution, and demineralized in AFS (acetic acid, formaldehyde, sodium chloride). Mandibles were processed and embedded in paraffin. Serial sections of 5 μm thickness were collected in the coronal (buccal-lingual) plane. Sections were stained for tartrate-resistant acid phosphatase (TRAP) to examine osteoclast activity and numbers (Sigma-Aldrich Kit, St. Louis, MO, USA), and Fast Green counterstaining and examined with a Nikon Eclipse 90i Advanced Research Scope. Representative images (40x) were taken of the alveolar bone furcation.

## Microbiome analysis

### DNA extraction

Mandible samples were cryogrinded and homogenized using bead-beating tubes and ceramic beads. Bacterial genomic DNA was extracted using the QIAamp DNA Microbiome Kit (Qiagen, Hilden, Germany) and further purified and concentrated using DNA Clean and Concentrator Kit (Zymo Research, Irvine, CA, USA) according to the manufacturer's protocol, then stored at −80 ˚C until all samples were collected.

### Sequencing

The V3-V4 variable region of the 16 s ribosomal RNA gene was amplified using gene-specific primers with Illumina adapter overhang sequences (5'-TCGTCGGCAGCGTCAGATGTGTATAAGAGA-CAGCCTACGGGNGGCWGCAG-3' and 5'-GTCTCGTGGGCTCGGAGATGTGTATAAGAGACAG-GACTACHVGGGTATCTAATCC-3'). Each reaction mixture contained 2.5 μl of genomic DNA, 5 μl of each 1 μM primer, and 12.5 μl of KAPA HiFi HotStart ReadyMix. Amplicon PCR was carried out as

follows: denaturation at 95℃ for 3 min, 35–40 cycles at 95℃ for 30 s, 55℃ for 30 s, 72℃ for 30 s, followed by a final extension step at 72℃ for 5 min. PCR products were verified using gel electrophoresis (1% agarose gel) and cleaned with AMPure XP beads (Agencourt, Beckman Coulter Inc, Pasadena, CA, USA). Amplicons were then indexed using the Nextera XT Index Kit V2 set A and set D (Illumina) and purified again with AMPure XP beads to remove low molecular weight primers and primer-dimer sequences. DNA concentrations were concentration of 1–2 nM using the SequalPrep Normalization Kit (Invitrogen). Samples were pooled into a single library which was analyzed using the TapeStation 4200 High Sensitivity D1000 assay (Agilent Technologies, Waldbronn, Germany) and Qubit High Sensitivity dsDNA assay (Thermo Fischer Scientific) to assess DNA quality and quantity. The final pooled library was then loaded on to an Illumina MiSeq sequencer with 10% PhiX spike, which served as an internal control to balance for possible low diversity and base bias present in the 16S amplicon samples, and was run for 478 cycles and generated a total of 5.68 million paired-end reads (2 × 239 bp).

## Bioinformatics

Raw paired-end sequences were imported in to Qiime2 (v. 2019.1) and were trimmed by 15 nt from the 5' end and truncated to 239 nt for the 3' end for both the forward and reverse reads respectively. The trimmed reads were then demultiplexed and denoised using the DADA2 package (*Callahan et al., 2016*). Forward reads were only used in our analysis. Taxonomy was then assigned using the feature-classifier suite trained on the Human Oral Microbiome Database (HOMD v. 15.1) (*Escapa et al., 2018*). Samples were then filtered for taxonomic contaminants excluded samples with less than 10,000 reads. Alpha and Beta diversity as well as other analysis were done in R-Studio using the Phyloseq (*McMurdie and Holmes, 2013*) Clustvis (*Metsalu and Vilo, 2015*), ggplot2 (*Wikham, 2016*), ampvis2 (*Andersen KS et al., 2018*), vegan (*Oksanen J et al., 2019*), ade4 (*Bougeard and Dray, 2018*) packages as part of the R suite.

Taxonomy filtered from samples was determined by analysis of kit controls with no template and zymo sequencing controls of known diversity and abundance in the QIAamp DNA Microbiome Kit (Qiagen, Hilden, Germany) and the DNA Clean and Concentrator Kit (Zymo Research). The following taxonomic assignments were removed as part of the dada2 workflow (*Callahan et al., 2016*): Unassigned, Cyanobacteria, acidovorans, pestis, coli, flavescens, sakazakii, durans, diminuta, anthropi, monocytogenes, parasanquinis_clade_411, otitidis, subtilis, aeruginosa, fermentum.

## Statistical analysis

Results for μCT analysis, including measurements, quantitative histology, proteome analysis are expressed as mean ± standard error of mean (SEM). Data were analyzed where appropriate using Student's t-test or paired t-test (comparing two groups only), or one-way analysis of variance (ANOVA) with post-hoc Tukey test for multiple comparisons, where p-values<0.05 were considered statistically significant. Statistical analysis was completed GraphPad Prism 8.00 (Graphpad, Software, La Jolla, CA, USA). For the 16 s rRNA sequencing, to identify statistically significant differences among agglomerated and normalized amplicon sequence variants (ASV) between samples as well as differences in alpha and beta diversity measures, we applied both the unpaired Wilcoxon rank-sum test as well as the two-tailed paired t-test – both with a 95% confidence interval ($\alpha$ = 0.05). Alpha diversity was assessed measuring Shannon, Chao1, Observed (ASV), and Fisher diversity measures. Beta diversity was measured using weighted Unifrac distances. Statistical analysis for the microbiome analysis was completed in R (v. 3.5.3).

## Data and materials availability

All data used in the development of this manuscript and the supplemental material are available in the manuscript or the supplemental materials or upon request. Bioinformatic scripts and microbiome data used in the analysis and generation of figures for this manuscript are available on the McLean Lab GitHub repository: https://github.com/kkerns85/Rapamycin_rejuvenates_oral_health_in_aging_mice.git (*Kerns, 2020*; copy archived at https://github.com/elifesciences-publications/Rapamycin_rejuvenates_oral_health_in_aging_mice). In addition, a web version of the R Markdown is available on Rpubs: https://rpubs.com/kkerns85/Rapamycin_Rmrkdown. The V4-16S rDNA sequences in raw

format, prior to post-processing and data analysis, have been deposited at the European Nucleotide Archive (ENA) under study accession no. PRJEB35672.

## Acknowledgements

The authors would like to thank the Karel F Liem Imaging Facility at Friday Harbor Laboratories, and Ryan Anderson on his guidance at Seattle Children's Research Institute. Authors would also like to thank Ella Lamont and Archita Gadkari for their guidance and processing of the microbiome samples.

## Additional information

### Competing interests

Matt Kaeberlein: Reviewing editor, *eLife*. The other authors declare that no competing interests exist.

### Funding

| Funder | Grant reference number | Author |
| --- | --- | --- |
| National Institute of Dental and Craniofacial Research | DE027254 | Jonathan Y An |
| National Institute of Dental and Craniofacial Research | DE023810 | Jonathan Y An Jeffrey S McLean |
| National Institute of Dental and Craniofacial Research | DE020102 | Jonathan Y An Jeffrey S McLean |
| National Institute on Aging | AG054180 | Catherine Kaczorowski |
| National Institute on Aging | AG038070 | Catherine Kaczorowski |
| National Institute on Aging | AG038070 | Catherine Kaczorowski |
| National Institutes of Health | TR002318 | Kristopher A Kerns |
| National Institute on Aging | AG013280 | Matt Kaeberlein |

The funders had no role in study design, data collection and interpretation, or the decision to submit the work for publication.

### Author contributions

Jonathan Y An, Conceptualization, Investigation, Methodology, Writing - original draft, Writing - review and editing; Kristopher A Kerns, Formal analysis, Investigation, Methodology; Andrew Ouellette, Laura Robinson, So-Il Park, Title Mekvanich, Alex Kang, Investigation; H Douglas Morris, Investigation, Methodology, Writing - review and editing; Catherine Kaczorowski, Resources, Supervision, Methodology, Writing - review and editing; Jeffrey S McLean, Resources, Formal analysis, Supervision, Methodology, Writing - review and editing; Timothy C Cox, Resources, Software, Methodology; Matt Kaeberlein, Conceptualization, Resources, Supervision, Visualization, Writing - original draft, Writing - review and editing

### Author ORCIDs

Jonathan Y An (iD) https://orcid.org/0000-0001-8422-8608
Kristopher A Kerns (iD) https://orcid.org/0000-0002-4380-0062
H Douglas Morris (iD) http://orcid.org/0000-0002-7942-3748
Jeffrey S McLean (iD) https://orcid.org/0000-0001-9934-5137
Matt Kaeberlein (iD) https://orcid.org/0000-0002-1311-3421

### Ethics

Animal experimentation: This study was performed in strict accordance with the recommendations in the Guide for the Care and Use of Laboratory Animals of the National Institutes of Health. All of

the animals were handled according to approved institutional animal care and use committee (IACUC) protocols of the University of Washington (#4359-01) and of the Jackson Laboratory (#06005-A24).

## Decision letter and Author response
Decision letter https://doi.org/10.7554/eLife.54318.sa1
Author response https://doi.org/10.7554/eLife.54318.sa2

## Additional files

### Supplementary files
• Transparent reporting form

### Data availability
The V4-16S rDNA sequences in raw format, prior to post-processing and data analysis, have been deposited at the European Nucleotide Archive (ENA) under study accession no. PRJEB35672. Dryad Data link: https://doi.org/10.5061/dryad.f4qrfj6sn.

The following datasets were generated:

| Author(s) | Year | Dataset title | Dataset URL | Database and Identifier |
|---|---|---|---|---|
| An JY, Kerns KA, Ouellette A, Robinson L, Morris D, Kaczorowski C, Park S, Mekvanich T, Kang A, McLean JS, Cox TC, Kaeberlein M | 2020 | Rapamycin rejuvenates oral health in aging mice | http://dx.doi.org/10.5061/dryad.f4qrfj6sn | Dryad Digital Repository, 10.5061/dryad.f4qrfj6sn |
| An JY, Kerns KA, Ouellette A, Robinson L, Morris D, Kaczorowski C, Park S, Mekvanich T, Kang A, McLean JS, Cox TC, Kaeberlein M | 2019 | Rapamycin rejuvenates oral health in aging mice | https://www.ebi.ac.uk/ena/browser/view/PRJEB35672 | European Nucleotide Archive (ENA), PRJEB35672 |

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
