## [Decision Letter]

Thank you for submitting your work entitled "Rapamycin rejuvenates oral health in aging mice" for consideration by *eLife*. Your article has been reviewed by two peer reviewers, one of whom is a member of our Board of Reviewing Editors, and the evaluation has been overseen by Jessica Tyler as the Senior Editor.

The manuscript has been improved but there are some remaining issues that need to be addressed as outlined below:

Reviewer #1:

This manuscript reports studies focused on the role of mTOR activity on age-associated periodontal bone loss and gingival inflammation, and the impact of mTOR attenuation with rapamycin on these features of oral cavity aging. The data presented are robust, provide important insights into the mechanisms of oral cavity aging and have substantial translational implications. No major deficiencies are noted.

Reviewer #2:

In this study, An and colleagues examine the effect of a relatively short-term (~2 months) rapamycin treatment on the periodontal health of aged mice. To assess periodontal health, the authors use complementary methods including microCT and microbiome analysis using 16S sequencing. The authors show that short-term rapamycin treatment improves a number of periodontal health parameters, and may thus have previously uncharted benefits in this aspect of age-related decline. The study uses 2 cohorts of mice at 2 independent sites (JAX and UW), which is unusual and very much strengthens the conclusions.

I have a few comments that I believe can be mostly addressed textually, which will improve the clarity of the report, and help improve future reproducibility of analysis.

1) The authors mention "animals" at both sites but do not specify the sex of the animals. Since a number of age-related phenotypes, including some rapamycin-related phenotypes, have been shown to be sex-dimorphic, it is critical to specify which sex was used for each cohort, at least in the Materials and methods section.

2) In this study, rapamycin is delivered through supplementation of diet, and the authors mention that the control chow is reinforced with "Eudragit". This reviewer thus assumes that this is to address potential concerns with regular chow diet as a control? It would be extremely helpful to explain the choice of Eudragit (as opposed to normal chow diet) in the main text or Materials and methods.

3) In their microbiome analysis, the authors mention significance between old control vs. treated (with regards to Figure 5), and no significance between young and old treated. What about young vs. old? Some of the rapa-induced changes look like they may "overcorrect", which should be explicitly discussed in the text as a potential caveat.

4) The authors show both microbiome analysis from combined JAX and UW mice in the main figures and a separated analysis in the supplement. Since there are visible specificities in the supplement based on cohort, did the authors have to apply batch-correction of any sort? It would be great to specify analytical parameters, including batch inclusion (or why it was not warranted) in the Materials and methods.

5) Could the authors speculate in their Discussion on whether the observed effects are expected to be maintained after the end of the short-term treatment, or if repeated treatments would be required?

---

## [Author Response]

Reviewer #2:[…] I have a few comments that I believe can be mostly addressed textually, which will improve the clarity of the report, and help improve future reproducibility of analysis.1) The authors mention "animals" at both sites but do not specify the sex of the animals. Since a number of age-related phenotypes, including some rapamycin-related phenotypes, have been shown to be sex-dimorphic, it is critical to specify which sex was used for each cohort, at least in the Materials and methods section.

This has now been clarified in the Materials and methods section. All of the mice were female.

2) In this study, rapamycin is delivered through supplementation of diet, and the authors mention that the control chow is reinforced with "Eudragit". This reviewer thus assumes that this is to address potential concerns with regular chow diet as a control? It would be extremely helpful to explain the choice of Eudragit (as opposed to normal chow diet) in the main text or Materials and methods.

This has been addressed in Materials and methods section. Eudragit is the encapsulation material used in eRAPA and is a copolymer derived from esters of acrylic and methacrylic acids. Eudragit without rapamycin was thus added to the regular chow at 42 ppm as a vehicle control.

3) In their microbiome analysis, the authors mention significance between old control vs. treated (with regards to Figure 5), and no significance between young and old treated. What about young vs. old? Some of the rapa-induced changes look like they may "overcorrect", which should be explicitly discussed in the text as a potential caveat.

We have worked to clarify this. Significant differences were observed between young and old for α diversity (Figure 5A), Firmicutes (Figure 5B) and Proteobacteria (Figure 5B). The potential overcompensation by rapamycin-induced changes such as the trend of apparent lower Bacteroidetes in the rapamycin-treated old vs. young is not statistically supported but could be a real effect. This is now mentioned explicitly in the text.

4) The authors show both microbiome analysis from combined JAX and UW mice in the main figures and a separated analysis in the supplement. Since there are visible specificities in the supplement based on cohort, did the authors have to apply batch-correction of any sort? It would be great to specify analytical parameters, including batch inclusion (or why it was not warranted) in the Materials and methods.

We agree that there were differences by cohort, which is not unexpected. Possible sources of these differences include the facilities and base diet formulations. All samples for microbiome and other analyses were processed in the same lab by the same person who was blinded to the conditions. DNA library preparation and sequencing were conducted in one run. We recognize that there may be species and strain level differences across sites, however at a higher taxonomic level, the differences which were seen in each site independently, did not result in higher variation leading to less significant differences overall. In terms of the applying batch-correction, we thank you for pointing out this concern to us. We concluded that there was no batch effect observed within this data analysis. We have included a Principle Coordinate Analysis (PCoA) using weighted Unifrac distances as Figure 5—figure supplement 4 which shows no clear separation based on either site location and/or group designation. Furthermore, a PERMANOVA analysis using 999 permutations was performed with Bray Curtis calculated distances using the Vegan (2.5-6) Package within the R suite. Results from this PERMANOVA analysis for site location and group designation indicated that no significant differences were observed based on these metadata variables (p = 0.341, ns).

5) Could the authors speculate in their Discussion on whether the observed effects are expected to be maintained after the end of the short-term treatment, or if repeated treatments would be required?

This is definitely an exciting avenue we are currently investigating. We have addressed this in the Discussion section of the main text and noted the published results from Quarles et al. showing that cardiac effects of rapamycin persist for at least 8 weeks following cessation of treatment.